# Hypoxia Sensing and Responses in Parkinson’s Disease

**DOI:** 10.3390/ijms25031759

**Published:** 2024-02-01

**Authors:** Johannes Burtscher, Yves Duderstadt, Hannes Gatterer, Martin Burtscher, Roman Vozdek, Grégoire P. Millet, Andrew A. Hicks, Hannelore Ehrenreich, Martin Kopp

**Affiliations:** 1Institute of Sport Sciences, University of Lausanne, 1015 Lausanne, Switzerland; gregoire.millet@unil.ch; 2Division of Cardiology and Angiology, University Hospital Magdeburg, 39120 Magdeburg, Germany; yves.duderstadt@med.ovgu.de; 3Research Group Neuroprotection, German Center for Neurodegenerative Diseases (DZNE), 39120 Magdeburg, Germany; 4Department of Sports Science, Otto-von-Guericke University, 39120 Magdeburg, Germany; 5Institute of Mountain Emergency Medicine, Eurac Research, 39100 Bolzano, Italy; hannes.gatterer@eurac.edu; 6Department of Sport Science, University of Innsbruck, 6020 Innsbruck, Austria; martin.burtscher@uibk.ac.at (M.B.); martin.kopp@uibk.ac.at (M.K.); 7Institute for Biomedicine, Eurac Research, Via Alessandro Volta 21, 39100 Bolzano, Italy; roman.vozdek@eurac.edu (R.V.); andrew.hicks@eurac.edu (A.A.H.); 8Clinical Neuroscience, Max Planck Institute for Multidisciplinary Sciences, 37075 Goettingen, Germany; hannelore.ehrenreich@web.de; 9Experimental Medicine, Central Institute of Mental Health, Medical Faculty Mannheim, Heidelberg University, 68159 Mannheim, Germany

**Keywords:** Parkinson’s disease, neurodegeneration, hypoxia, mitochondria, synuclein, respiratory diseases

## Abstract

Parkinson’s disease (PD) is associated with various deficits in sensing and responding to reductions in oxygen availability (hypoxia). Here we summarize the evidence pointing to a central role of hypoxia in PD, discuss the relation of hypoxia and oxygen dependence with pathological hallmarks of PD, including mitochondrial dysfunction, dopaminergic vulnerability, and alpha-synuclein-related pathology, and highlight the link with cellular and systemic oxygen sensing. We describe cases suggesting that hypoxia may trigger Parkinsonian symptoms but also emphasize that the endogenous systems that protect from hypoxia can be harnessed to protect from PD. Finally, we provide examples of preclinical and clinical research substantiating this potential.

## 1. Introduction

Being diagnosed with Parkinson’s disease (PD) means having to deal with a multisymptomatic, debilitating neurological disorder. At the time of diagnosis, the molecular disease mechanisms progress inexorably, and pronounced degeneration of dopaminergic neurons in the *substantia nigra pars compacta* has already occurred, leading to compromised dopaminergic innervation of the basal ganglia (nigrostriatal dopaminergic pathway) and motor symptoms, bradykinesia (slow movements), rigidity, resting tremor, impaired postural control, and gait deficits [1,2]. More than 90% of cases cannot be clearly attributed to genetic causes and are thus considered sporadic. Age is the main risk factor in these cases [3], contributing to the rapid increase in the global prevalence of PD in parallel to increasing life expectancies. Despite major scientific advances culminating in spectacular successes in preclinical models, no disease-modifying treatments are yet available [4]. The mainstay pharmacological treatments focus on the re-establishment of dopaminergic signaling. They have been in use for several decades and include the administration of dopamine precursors (L-DOPA) and dopamine receptor agonists [5]. While initially these approaches frequently reduce some motor symptoms, efficiency tends to wear off with time, and L-DOPA can induce dyskinesia by itself, in particular in later disease stages (levodopa-induced dyskinesia) [6]. In addition, several devastating symptoms, particularly non-motor symptoms such as fatigue, sleep disturbances, anxiety, depression, or impaired autonomic regulation cannot be effectively treated. However, these symptoms are important predictors of health-related quality of life in PD [7]. In addition to pharmacological strategies, deep-brain stimulation (often of the subthalamic nucleus and thereby dopaminergic neuronal networks) has emerged as a therapy of choice for a subgroup of patients. Targeting other pathological hallmarks like mitochondrial dysfunction and aberrant aggregation of the protein alpha-synuclein, resulting in characteristic Lewy pathology, was successful in animal models of PD, but until now, not in clinical trials [8]. Still, the debated FDA approval for the amyloid-beta antibodies aducanumab and lecanemab for Alzheimer’s disease indicates a general trend towards approaches aimed at reducing neurodegeneration-related protein aggregations [9]. A summary of approved treatment strategies and important trends in the development of new strategies can be found, for example, in a recent review by Chopade and colleagues [10]. Briefly, besides the booming anti-aggregation strategies, neurotransmitter-system modulating (including variations of levodopa/carbidopa treatments or, e.g., subcutaneous application of apomorphine), gene therapy (e.g., viral delivery of glial-derived neurotrophic factor genes), anti-inflammatory (e.g., the diabetes-approved drug Exenatide or omega-3 fatty acids), anti-oxidant (e.g., oral cannabidiol), or regenerative strategies (e.g., to promote dopaminergic neurogenesis) are being tested in clinical trials.

Taken together, the failure to translate preclinical findings into efficient clinical therapies highlights the need to better understand the causal and contributing factors underlying PD development and progression. While dopaminergic neurodegeneration accounts for PD’s cardinal motor symptoms, the reason for the death of these cells remains unclear. Although alpha-synuclein pathology, mitochondrial dysfunctions, and other factors are probably involved, and mutations of genes linked to those pathogenic factors can result in familial PD [11,12], a complex interplay of genetic susceptibility and environmental contributions is thought to cause sporadic PD. Here we suggest that systemic and local oxygen deficits and maladaptation to insufficient oxygen availability (hypoxia) may trigger a cascade of pathological processes that can be observed in PD and thus could contribute to disease onset and progression, at least in a subset of PD cases. We propose that the restoration of oxygen sensing capacity and physiological hypoxia responses and adaptations may be efficient strategies in PD, a hypothesis that requires experimental confirmation.

## 2. Severe Hypoxia Triggers Parkinsonian Symptoms

Hypoxia may directly trigger the development of Parkinsonian symptoms in vulnerable individuals, as suggested by a few case studies. Gait disturbances, tremor, limb rigidity, and bradykinesia were, for example, observed in a 64-year-old man who climbed a mountain of 2700 m above sea level (asl) for the first time in his life, but all symptoms resolved after several months [13]. The authors speculated that hypometabolism in the basal ganglia might have been the cause, but this hypothesis was not investigated further. Another case report describes the development of transient Parkinsonism in the absence of detected structural brain damage in a 56-year-old man; the authors suggested that this was the consequence of climbing a mountain of 3500 m asl, although the symptoms were recorded only 1 month after the altitude exposure [14]. An acute hypoxic insult during exposure to an altitude of 4876 m asl that led to unconsciousness resulted in the occurrence of Parkinsonism and emotional symptoms [15]. In this case, lesions of the globus pallidus in the basal ganglia were observed, and the motor symptoms improved with carbidopa/levodopa treatment. Most Parkinsonian signs were not observed any more 13 months after the hypoxic insult. A case of atypical Parkinsonism was also described in an individual who, from age 67, resided for 2 years in a research facility in which oxygen levels gradually declined to 14.2% [16].

Although the development of Parkinsonism after exposure to environmental (hypobaric) hypoxia has been reported only very rarely, these observations suggest that severe (as in very high altitudes, >3500 m asl, corresponding to a reduced inspired fraction of oxygen F_I_O_2_ of <14% versus a normoxic F_I_O_2_ of about 20.9%) and very rarely also moderate (as in high altitude, between 1500 and 3500 m asl, corresponding to an F_I_O_2_ of 18–14%) hypoxia can result in Parkinsonism or possibly trigger the development of PD in some vulnerable individuals. Such vulnerabilities require elucidation but may be linked to cardio-respiratory or genetic/molecular risk factors, as discussed below. An anatomical basis may be the well-established structural and functional vulnerability of basal ganglia circuits (including the striatum) to hypoxia, due in part to low perfusion despite a strong dependence on aerobic energy metabolism [17,18]. However, many other factors may have contributed to the development of symptoms in the reported case studies, e.g., climatic conditions, physical exertion, or stress related to high-altitude environments.

In contrast, a recent review concludes that exposure to both acute hypoxia and prolonged or repeated moderate hypoxia is generally safe for people with early or moderate PD [19]. Importantly, however, the hypoxic dose (intensity, duration, frequency) was not indicated in all analyzed studies but determined the safety of hypoxia together with individual vulnerabilities [20].

Another poorly understood link between hypoxia and PD may be sleep impairments, common prodromal features in PD [21] that can be associated with periods of severe hypoxia (sleep apnea). Indeed, a recent study found an increased risk for individuals suffering from obstructive sleep apnea to develop PD [22].

## 3. Impaired Systemic Oxygen-Sensing and Ventilatory Dysfunction

Severe hypoxia is a life-threatening condition, with several brain regions, including nigrostriatal networks, being particularly vulnerable to low oxygen availability. Thus, numerous molecular and systemic responses are in place to protect cells and the organism from hypoxic insult. On the systemic level, the peripheral oxygen sensors, primarily those of the carotid body, sense reduced oxygen tension and trigger an upregulation of ventilation (hypoxic ventilatory response (HVR)) and modulation of cardiovascular parameters (including elevated heart rate and cardiac output) to compensate for reduced oxygen availability [23].

In 1998, Serebrovskaja and colleagues published findings showing a reduced HVR in people with PD (only men were investigated); specifically, alveolar ventilation in severe hypoxia was 2.6 (sitting) to 4.6 (lying) times higher in healthy, age-matched men, despite the maintained mechanical function of the lung in PD [24]. Two years later, Onondera and colleagues reported impaired chemosensitivity to hypoxia and the perception of dyspnea (shortness of breath) underlying the blunted HVR already in early-stage PD [25].

Although many individuals with PD suffer from ventilatory dysfunction, the role of such symptoms in PD, e.g., whether they are causes or consequences of other PD-related pathologies and symptoms, is still poorly understood [26,27]. There are, however, indications for a significantly higher risk for people with respiratory diseases (e.g., chronic obstructive pulmonary disease (COPD)) [28] or obstructive sleep apnea [29]) to develop PD. In addition, people with PD and COPD comorbidities are at greater risk for PD-related hospitalization [30], potentially indicating a worse disease progression in this subpopulation. Similarly, stroke and PD appear to both increase the risk of each other, with a pooled odds ratio of PD postmortem brains exhibiting stroke pathology amounting to 1.86, as assessed by a recent meta-analysis [31]. Although causality remains to be established, this could indicate that diseases or events promoting hypoxia might facilitate or accelerate the development of PD.

Together with impaired autonomic cardiovascular regulation and systemic perfusion deficits, the ventilatory symptoms affect quality of life and autonomy but also present an additional obstacle to performing exercise [32]. This is especially problematic since different types of exercise have been demonstrated to robustly benefit people with PD, ameliorating motor and non-motor symptoms [33,34,35].

## 4. Hypoxia in the *Substantia Nigra* and Dopaminergic Signaling

Like the basal ganglia nuclei innervated by the *substantia nigra*, the dopaminergic neurons in the *substantia nigra* themselves are also highly vulnerable to hypoxia [36]. Nigral dopaminergic projection neurons innervate the striatum through long and branched axons and require great amounts of oxygen and ATP to satisfy their high energy demand, originating in part from their signaling activity [37,38]. They are, therefore, at particular risk for oxidative stress and insufficient oxygen supply. This risk is further exacerbated by the oxygen dependence of dopamine biosynthesis and metabolism [39], in particular of the rate-limiting enzyme tyrosine hydroxylase (TH) and of various catabolic pathways, including the action of monoamine oxidases (MAOs). TH converts tyrosine to the dopamine precursor DOPA, requiring ferrous iron, the cofactor tetrahydrobiopterin, and oxygen, while MAO requires oxygen to deaminate dopamine. The oxygen dependence of dopamine metabolism also represents an intrinsic risk for oxidative stress and necessitates elaborate antioxidative systems [40]. While mitochondrial membrane-bound MAO has recently been demonstrated to normally shuttle electrons resulting from the deamination of dopamine into the electron transport system to sustain ATP production and possibly protect dopamine from auto-oxidation [41], it is possible that pathological conditions or (hypoxic) insults disrupt the delicate redox balance in dopaminergic systems, leading to excessive reactive oxygen species (ROS) production. The vulnerability of the *substantia nigra* to hypoxia/ischemia has also been demonstrated in rat brain slices, in which oxygen–glucose deprivation elicited a nigral spreading depression, a spreading neuronal and glial depolarization phenomenon characteristic of ischemic-damage-susceptible brain regions [42].

In light of the danger of hypoxia, it is unsurprising that cells are equipped with elaborate pathways to sense and respond to hypoxia [43]. Hypoxia-inducible factors (HIF1 and 2) are integral components of these mechanisms. While being continuously degraded in normoxia (via prolyl hydroxylases or factor-inhibiting HIF [43]), HIF alpha-subunits are stabilized in hypoxia (due to reduced activities of oxygen-dependent prolyl hydroxylases and factor-inhibiting HIF), allowing them to dimerize with beta-subunits and orchestrate cellular responses to hypoxia. Notably, these include the upregulation of anaerobic energy production (glycolysis), modulation of mitochondrial activities, and promotion of oxygen supply, e.g., by upregulation of erythropoietin (EPO) and vascular endothelial growth factor (VEGF). Of relevance for PD may also be the control of TH by HIF [44,45], a surprisingly poorly investigated topic since dopamine levels are modulated by exposure to hypoxia. In a study conducted by Serebrovskaya and colleagues [46], for example, blood DOPA and dopamine levels increased significantly in older people (61 ± 1.4 years), and dopamine also increased in young healthy adults upon exposure to an altitude of 2200 m asl.

Several findings suggest that oxygen sensing or the associated cellular responses related to HIFs are impaired in PD. Accumulations of the hypoxia marker HIF2-alpha were detected in a subsample of PD post-mortem brains [47], and polymorphisms in HIF1 may be associated with PD [48]. In addition, the manipulation of HIF levels/stability may modulate PD symptomatology or progression, as discussed below. HIF signaling has therefore emerged as a promising target in PD [49] but also in other neurodegenerative diseases, such as Alzheimer’s disease [50].

An additional link between oxygen availability and dopaminergic signaling revolves around brain-derived neurotrophic factor (BDNF). BDNF is a fundamental modulator of neuronal processes and modulates brain functions by regulating synaptic plasticity, neuronal survival/development, and differentiation [51]. Reduced BDNF levels have been reported in many psychiatric and neurodegenerative diseases, such as major depressive disorder, Alzheimer’s disease, or PD [52,53]. BDNF’s preferential receptor, tropomyosin kinase B receptor (TrkB), is widely distributed in the central nervous system, including dopaminergic neurons of the *substantia nigra*, and increasing BDNF and/or TrkB expression is considered beneficial in PD [54,55]. BDNF levels are upregulated in response to exercise [56], and several studies suggest that short-term, moderate, controlled hypoxia increases BDNF levels as well, as shown in human pulmonary artery smooth muscle and endothelial cells [57,58], spinal cord [59], and brain stem [60] of rodents. Hypoxia, similar to exercise, may modulate PD pathology and symptomatology through effects on the BDNF system. However, the underlying neurobiological mechanisms linking hypoxia and BDNF are still poorly understood and require further research.

## 5. Alpha-Synuclein Aggregation and Hypoxia

Another central pathological aspect of PD is the misfolding and aggregation of alpha-synuclein. Interestingly, alpha-synuclein levels are known to increase in response to hypoxia, as demonstrated in cellular models [61] and mouse brain [62]. In line with these findings, obstructive sleep apnea, which is linked to hypoxic episodes during sleep, is associated with increased (circulating) alpha-synuclein levels in humans [63]. In addition, increased alpha-synuclein levels, as a consequence of hypoxia/ischemia likely contribute to the protein’s aggregation [64], and in cells, hypoxia leads to the formation of toxic alpha-synuclein oligomers [61]. Further substantiating the link between hypoxia/ischemia and PD, a recent study demonstrated that the induction of transient focal ischemia by middle cerebral artery occlusion in PD-model mice (expressing human alpha-synuclein with a known mutation leading to familial PD; A53T) led to progressive alpha-synuclein pathology and ultimately, the loss of dopaminergic neurons in the *substantia nigra* [65]. The link between hypoxia and alpha-synuclein-related pathology has recently been extensively reviewed [66,67], and the interactions with mitochondria appear to be central to pathological changes in response to hypoxia. Specifically, hypoxic cellular environments may alter mitochondrial membrane structures and the interaction with alpha-synuclein, thereby potentially facilitating Lewy pathology development and mitochondrial dysfunction.

## 6. Mitochondria

Mitochondria consume most of the cellular oxygen during oxidative phosphorylation, in which electrons are transferred to oxygen by cytochrome c oxidase, enabling ATP production. Consequently, hypoxia impairs mitochondrial activities and energy production, leading to increased production of ROS and oxidative stress, as well as energy deficiency and changes in mitochondrial dynamics, quality control (including the targeted clearance of damaged mitochondria by mitophagy), and biogenesis [68]. Mitochondrial dysfunction is strongly linked to PD, with impaired mitochondrial respiration observed in different tissues from the *substantia nigra* to platelets and skeletal muscle, and mutations of related genes (e.g., those expressing the mitophagy-related proteins PINK1 and parkin) are causes or risk factors for PD [12]. Moreover, pharmacological inhibition of mitochondrial respiration can cause parkinsonian symptoms, first described for 1-methyl-4-phenyl-1,2,3,6-tetrahydropyridine (MPTP) abuse [69], a mitochondrial complex I inhibitor that, since then, has been used as an inducer of PD pathology and symptoms in animal models [70]. Inhibitors of mitochondrial respiration in environmental toxins may also represent risk factors for PD, although clear evidence for this association is lacking [71].

Mitochondrial responses to hypoxia and the associated acidification of the cellular environment also affect alpha-synuclein pathology formation [36,66], and mitochondrial damage can result in cell death, contributing to the degeneration of vulnerable neurons in PD. Interestingly, hypoxia is associated with the upregulation of mitophagy via modulation of several molecules, including the mitochondrial outer membrane protein FUN14 domain containing 1 (FUNDC1), BCL2/adenovirus E1B 19-kDa-interacting protein 3 (BNIP3), and NIX/BNIP3L [72]. In addition, hypoxia may promote other mechanisms leading to mitochondrial degradation, such as the promotion of the interaction of mitochondria with lysosomes [73].

Overall, mitochondria are highly sensitive to oxygen deficiencies, and cells that depend most on adequate mitochondrial functioning, such as dopaminergic projection neurons of the *substantia nigra*, are vulnerable to the consequences of hypoxia.

## 7. Increasing the Resilience to Hypoxia in PD

The mechanisms described above all suggest an involvement of hypoxia and/or impaired responses to hypoxia as a crucial factor in PD development and progression. It is therefore plausible that improving these responses may be beneficial in PD, and there are indeed several lines of evidence suggesting this.

First, in various cellular and animal models of PD (toxin and α-synuclein-overexpression models), pharmacological strategies that enhance HIF stability/activity (e.g., using HIF-inducing iron chelators like deferoxamine, the HIF-activator agmantin, prolyl hydroxylase inhibitors like FG-4592, or the HIF-1α-transcription-enhancing compound albendazole) have been shown to be beneficial [74,75,76,77,78,79,80]. Second, responses to hypoxia can be improved by activation of delta opioid receptors, which have been shown to blunt alpha-synuclein expression and toxicity in a cellular toxin model of PD [81]. Third, hypoxia can be used to differentiate multipotent mesenchymal stem cells into dopaminergic neurons, which has been suggested to be useful for cell transplantation approaches in PD [82], but might also partly explain the benefits of hypoxia exposure in people with PD. Fourth, there are some indications that prolonged or repeated exposure to moderate hypoxia may improve motor (hypokinesia) and non-motor functions (including HVR) and increase quality of life, as recently reviewed [19]. Maybe the most promising evidence comes from the use of hypoxia-conditioning approaches, which aim at improving the cellular and systemic resilience to hypoxic insults [23]. Such strategies in humans commonly involve sessions (often about 30 min long) with several consecutive short periods of hypoxia (2–5 min F_I_O_2_ > 10%), interspersed with normoxic (or hyperoxic) resting phases of similar duration. Those sessions are repeated several times per week over usually 3–8 weeks and are referred to as intermittent hypoxia conditioning (IHC). Similar IHC protocols have recently been demonstrated to improve cognitive functions in older people with and without mild cognitive impairment [83], and thus present promising approaches for age-related neurological diseases. Moreover, IHC has been shown to improve the hypoxic ventilatory response in people with PD, especially if treated with L-DOPA (an English summary of results published in Russian can be found in [84]). Belikova and colleagues [85] demonstrated that a hypoxia conditioning protocol for rats (5 cycles of 15 min exposures to F_I_O_2_ = 12%) improved striatal dopamine availability and decreased lipid peroxidation. Since then, and in combination with the successes in other neurodegenerative diseases, hypoxia conditioning has been repeatedly proposed as a potential treatment strategy in humans with PD [86,87]. Recently, a clinical trial has been launched, investigating the impact of acute hypoxia exposure on people with PD [87]. Importantly, in this trial, hypoxia exposures for 45 min or for 5 × 5 min (both with F_I_O_2_ of 12.7, or 16.3%) are applied in single sessions. While the results of these studies will provide important information on individual acute responses of people with PD to hypoxia, the potential long-term benefits, as observed in 3–8-week hypoxia conditioning protocols [83], will remain undiscovered.

Apart from the potential of hypoxia conditioning to improve brain health, in particular, its demonstrated efficiency in respiratory diseases and autonomic impairments merits consideration. Hypoxia conditioning has a long history of efficient application in chronic obstructive pulmonary disease (COPD), where, among other benefits, it improves ventilation [88]. Furthermore, in COPD patients, hypoxia interventions improved autonomic cardiovascular dysfunctions (e.g., baroreflex sensitivity) [89], which are also common features in PD [90].

Moreover, future research should also focus on the extent to which IHC protocols could address other non-motor symptoms of PD (fatigue, anxiety, or depression) and health-related quality of life. In the same respect as for the molecular and cardiorespiratory consequences of hypoxia, hypoxia seems to also allow a bidirectional way of thinking with regard to stress-related pathways: exerting detrimental effects at high doses and beneficial effects at appropriately chosen lower doses [91]. This may be especially relevant since chronic stress-related hormonal changes may accelerate PD pathology formation, as demonstrated in a mouse model of PD [92].

In summary, there is evidence that cellular and systemic natural defense mechanisms to hypoxia can be activated both by pharmacological (e.g., modulation of the HIF pathways) and non-pharmacological ways (e.g., controlled exposure to inspirational hypoxia). These defense mechanisms (see Figure 1) importantly include antioxidant and anti-inflammatory mechanisms and the induction of (largely HIF-mediated) protective mechanisms to maintain energy availability and cellular homeostasis by improvements in mitochondrial resilience, increased flexibility of substrate utilization, switching to alternative metabolic pathways (primarily glycolysis), and the optimization of oxygen delivery, as well as by improving cellular waste disposal [23,43], all neuroprotective mechanisms that may counteract PD neuropathology. Conversely, uncontrolled (both continuous and intermittent) hypoxia is detrimental for cells and organs if the activation of the defense mechanisms is insufficient, either because the hypoxia is too severe, or the defense capacities are too weak. Regarding waste disposal, the unfolded protein response [93], chaperone-mediated autophagy [94,95], and mitophagy [72] are all implicated in PD pathology and are modulated by hypoxic stress.

## 8. Conclusions and Perspectives

Reduced oxygen availability is a major threat for the brain, but the view that hypoxia is always negative is too general. Rather, the hypoxic dose and individual or disease-related vulnerabilities determine if hypoxia has beneficial or detrimental effects under specific circumstances. Recently, life-long chronic continuous hypoxia (F_I_O_2_ of 11%) has been demonstrated to protect the brain in animal models of neurological diseases such as Leigh syndrome [96,97] or Friedreich ataxia [98], presumably by correcting the disease-induced increased levels of oxygen in the brain, which may promote oxidative stress. Furthermore, living at higher altitudes is associated with improved function and structure of the neurovascular unit in populations such as Andean highlanders [99] and a reduced risk of stroke in moderate-altitude residents in the Alps [100]. Chronic, mild, and continuous hypoxia may therefore protect the brain by optimizing cardiovascular parameters. In a different approach, mild or brief episodes of hypoxia can be applied to induce beneficial responses both on the cellular and systemic level [23], which improve metabolic flexibility (higher efficiency of oxidative metabolism and switching to anaerobic pathways) and the induction of anti-oxidative and anti-inflammatory defense mechanisms, as well as protective respiratory and cardiovascular responses [23]. These effects of inspiratory hypoxia likely overlap with and may, to some degree, reproduce or amplify so-called “functional hypoxia” caused by increased oxygen demand, for example, in skeletal muscle during exercise [101] or in the brain due to motor-cognitive training [102,103,104]. Therefore, “functional hypoxia” may also play a role in the well-documented benefits of exercise [35] and motor–cognitive training [105] in PD.

Based on the important deficits related to sensing and responding to hypoxia in PD, we suggest that the inability to deal with hypoxic challenges (due to either very severe hypoxia or due to insufficient molecular or systemic protective capacities) is a crucial factor in the pathogenesis or progression of neuropathology and symptomatology in PD. Consequently, the components involved in oxygen sensing and hypoxia responses may represent therapeutic targets (Figure 2).

In line with this reasoning, pharmacological strategies to increase HIF alpha-subunit stabilization are emerging as powerful interventional approaches in cell and animal models of PD, and the application of hypoxia conditioning in PD recently gained renewed interest, long after reports by Russian scientists on the effectiveness of this approach decades ago. Such treatments that improve cellular and systemic oxygen sensing and improve resilience to later hypoxic challenges/insults also have the potential to maintain mitochondrial function, prevent alpha-synuclein misfolding and aggregation, and preserve dopamine signaling. Moreover, they can improve the systemic sensitivity and thereby respiratory and cardiovascular responses to hypoxia, resulting in improved maintenance of sufficient systemic oxygen supply.

However, even though clinical trials are on the way and will provide important new information on individual responses to hypoxia in people with PD [87], we probably should not expect to be able to appreciate the full potential of hypoxia conditioning in PD too soon. The determination of the hypoxic doses required to induce an optimal effect while also minimizing hypoxia-associated risks will be a major future challenge, especially since many parameters have to be optimized, including the intensity, duration, and frequency of the hypoxic stimulus, but also the type of hypoxia (normobaric versus hypobaric), the role of CO_2_ levels (isocapnic versus poikilocapnic hyperventilation in hypoxia reduces CO_2_ levels), and various environmental (e.g., temperature) and behavioral (e.g., activity level) factors likely play a decisive role. Increased CO_2_ levels, by themselves or in combination with hypoxia (e.g., by adding CO_2_ to the breathing gas), for example, promote ventilatory plasticity [106] and increased cerebral blood flow [107]. While the evidence is mounting that responses to oxygen level changes are crucial in neurological diseases, and especially PD, understanding the role of the factor oxygen requires systematic investigation.

## Figures and Tables

**Figure 1 ijms-25-01759-f001:**
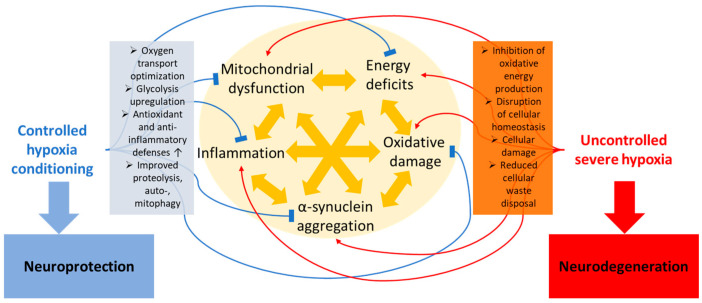
Suggested mechanisms induced by hypoxia leading to either neuroprotection or neurodegeneration. While severe hypoxia is detrimental if natural defense mechanisms fail, controlled hypoxia can be used to strengthen those cellular and systemic defense mechanisms by inducing adaptations leading to increased resilience.

**Figure 2 ijms-25-01759-f002:**
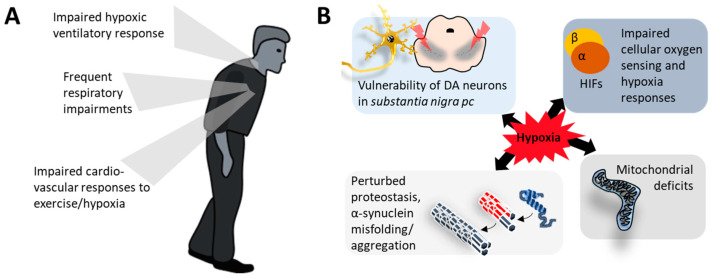
Systemic deficits in responses to changing oxygen levels in PD (**A**) and cellular consequences of hypoxia (**B**). An impaired capacity to sense blood oxygen levels and/or a blunted response to hypoxia (respiration, cardiovascular) may underly the vulnerability of people with PD to hypoxia. Several prominent molecular and cellular aspects of PD are related to hypoxia sensing and hypoxia responses. Among them are the exquisite vulnerability of the dopaminergic (DA) system in the *substantia nigra pars compacta* (pc), impaired functioning of hypoxia-inducible factor (HIF) signaling, mitochondrial deficits, and the sensitivity of alpha-synuclein expression and aggregation to oxygen levels.

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
