# Peer review of "Hypoxia Sensing and Responses in Parkinson’s Disease"

_ijms, 2024, doi:10.3390/ijms25031759_

Round 1

Reviewer 1 Report

Comments and Suggestions for Authors

The manuscript is dedicated to the analysis of current scientific data on the functions of role of hypoxia in signaling pathways of Parkinson's disease mechanisms. The review also includes information about endogenous systems that protect against hypoxia and can be used to protect against Parkinson's disease.

The study has been carried out at a high methodological level: more than 95 literature sources have been analyzed, most of which were published in the last 3-5 years.

Of particular interest is section 8, where conclusions are drawn about the high importance of the hypoxic component in the pathogenesis of Parkinson's disease and the possibility of using systemic resistance to hypoxia as a therapeutic target for this disease.

However, there are several recommendations that will help the authors improve the manuscript:

1.      1. The abbreviations of FUNC1 and BNIP3 are missing from the list of abbreviations. Check it again.

2.      2. There is work that also confirms the authors' conclusions and was performed in vitro using the OGD oxygen and glucose deprivation model): https://pubmed.ncbi.nlm.nih.gov/23796781 / . It should probably also be considered in the manuscript.

3.      3. In the body, tissues and cells, hypoxia is often accompanied by acidosis and excess CO2 (hypercapnia). At the same time, there is information about the potentiation of the adaptogenic effects of hypoxia when combined with an excess of carbon dioxide in tissues. What do the authors think, is it possible to have a positive effect of moderate hypercapnia as a therapeutic target in Parkinson's disease?

Author Response

Dear editor, dear reviewers,

We are very grateful for the very supportive and positive feedback. All suggestions have been integrated into the new, revised version of the manuscript, as outlined in the point-by-point replies below.

The revisions have been performed on the document provided on the MDPI website. Since the endnote links had been removed in this new version, new references were added in a different style for clarity and the full details are provided under the original bibliography (even in the clean version). We hope that this procedure will facilitate final editing.

Best regards,

Johannes Burtscher, for the authors

Reviewer 1

The manuscript is dedicated to the analysis of current scientific data on the functions of role of hypoxia in signaling pathways of Parkinson's disease mechanisms. The review also includes information about endogenous systems that protect against hypoxia and can be used to protect against Parkinson's disease.

The study has been carried out at a high methodological level: more than 95 literature sources have been analyzed, most of which were published in the last 3-5 years.

Of particular interest is section 8, where conclusions are drawn about the high importance of the hypoxic component in the pathogenesis of Parkinson's disease and the possibility of using systemic resistance to hypoxia as a therapeutic target for this disease.

However, there are several recommendations that will help the authors improve the manuscript:

Re. We thank the reviewer for the favorable evaluation of our manuscript and the valuable suggestions, which we fully addressed in the revised version of the review.

  1. The abbreviations of FUNC1 and BNIP3 are missing from the list of abbreviations. Check it again.

Re. Thank you for making us aware of this omission. These abbreviations have been added and the entire manuscript has been double-checked for consistency of abbreviations and completeness of the abbreviation registry.

  1. There is work that also confirms the authors' conclusions and was performed in vitro using the OGD oxygen and glucose deprivation model): https://pubmed.ncbi.nlm.nih.gov/23796781 / . It should probably also be considered in the manuscript.

Re. Thank you for pointing out this work, confirming the substantia nigra’s vulnerability to hypoxia. We added the following paragraph to chapter 4 (Hypoxia in the substantia nigra and dopaminergic signaling):

The vulnerability of the substantia nigra to hypoxia/ischemia has also been demon-strated in rat brain slices, in which oxygen-glucose deprivation elicited a nigral spreading depression, a spreading neuronal and glial depolarization phenomenon characteristic for ischemic-damage susceptible brain regions (Karunasinghe and Lipski, 2013).

  1. In the body, tissues and cells, hypoxia is often accompanied by acidosis and excess CO2 (hypercapnia). At the same time, there is information about the potentiation of the adaptogenic effects of hypoxia when combined with an excess of carbon dioxide in tissues. What do the authors think, is it possible to have a positive effect of moderate hypercapnia as a therapeutic target in Parkinson's disease?

Re. We thank the reviewer for raising this important question. Hypercapnia indeed appears to exert important effects on the brain and therefore likely is a crucial modulator of hypoxia-induced risks and benefits. For example, hypercapnia plays a role in promoting ventilatory plasticity next to hypoxia and enhances cerebral blood flow. Therefore, we agree that moderate hypercapnia may be beneficial in PD, probably especially in combination with controlled hypoxia and that this is an avenue that deserves research. We added to the CO2 part in the conclusions/perspectives:

Increased CO2-levels, by themselves or in combination with hypoxia (e.g., by adding CO2 to the breathing gas), for example promote ventilatory plasticity (Griffin et al., 2012) and increased cerebral blood flow (Ito et al., 2003).

Reviewer 2 Report

Comments and Suggestions for Authors

The review entitled “Hypoxia Sensing and Responses in Parkinson’s Disease” reveals an interesting role of hypoxic episodes in the development of Parkinson’s disease pathology. Due to the lack of effective therapy against Parkinson’s disease and limited information about the pathogenesis, researchers extensively study neurodegenerative disorders and the chronic diseases associated with their development. Therefore, the subject of the manuscript is extremely important and the authors describe the issues in a good manner. The manuscript is well composed and conclusion about the involvement of hypoxia in the development or treatment of pathology is based on many cited publications. I present only a few comments or questions below:

1. In the introduction there is information about animal models of Parkinson’s disease and about currently tested therapy. Could authors add short information about clinical trials and potential therapy tested in the context of human research system?

2. Are there any other situations during which oxygen deficiency occurs for example chronic obstructive pulmonary disease or other examples of chronic disorders? Are there any information about risk of Parkinson’s disease onset and aforementioned chronic disorders?

3. In line 197-199 authors write that alpha-synuclein levels are known to change in response to hypoxia. Could you please specify the direction of these changes?

4. Summing up can we assume that stroke may be the cause of the Parkinson’s disease onset? Is there any relationship between Parkinson’s disease onset/pathology and vascular risk factors, diabetes or hypertension?

5. Are there any natural defense mechanism against hypoxia that could be supported to prevent neurodegenerative disorders including Parkinson’s disease?

6. Paragraph 7: Could you precise what molecular mechanisms may underlie the beneficial hypoxia, preserving brain metabolism and neuroprotection? Could you support it with scheme presenting detrimental and beneficial roles of oxygen deficits?

Author Response

Dear editor, dear reviewers,

We are very grateful for the very supportive and positive feedback. All suggestions have been integrated into the new, revised version of the manuscript, as outlined in the point-by-point replies below.

The revisions have been performed on the document provided on the MDPI website. Since the endnote links had been removed in this new version, new references were added in a different style for clarity and the full details are provided under the original bibliography (even in the clean version). We hope that this procedure will facilitate final editing.

Best regards,

Johannes Burtscher, for the authors

Reviewer 2

The review entitled “Hypoxia Sensing and Responses in Parkinson’s Disease” reveals an interesting role of hypoxic episodes in the development of Parkinson’s disease pathology. Due to the lack of effective therapy against Parkinson’s disease and limited information about the pathogenesis, researchers extensively study neurodegenerative disorders and the chronic diseases associated with their development. Therefore, the subject of the manuscript is extremely important and the authors describe the issues in a good manner. The manuscript is well composed and conclusion about the involvement of hypoxia in the development or treatment of pathology is based on many cited publications. I present only a few comments or questions below:

Re. We are very grateful for the positive evaluation of our manuscript and for highlighting the importance of the topic. Please find a description of the performed revisions as suggested by the reviewer below.

  1. In the introduction there is information about animal models of Parkinson’s disease and about currently tested therapy. Could authors add short information about clinical trials and potential therapy tested in the context of human research system?

Re. Thank you for this excellent suggestion. We extended the paragraph on treatment strategies and added the following information:

Targeting other pathological hallmarks like mitochondrial dysfunction and aberrant aggregation of the protein alpha-synuclein, resulting in characteristic Lewy pathology, was successful in animal-models of PD but until now not in clinical trials; still the debated FDA-approval for the amyloid-beta antibodies aducanumab and lecanemab for Alzheimer’s disease indicates a general trend towards approaches aimed at reducing neurodegeneration-related protein aggregations (Høilund-Carlsen et al., 2023). A summary of approved treatment strategies and of important trends in the development of new strategies can be found for example in a recent review by Chopade and colleagues (Chopade et al., 2023). Briefly, besides the booming anti-aggregation strategies, neurotransmitter-system modulating (including variations of levodopa/carbidopa treatments or e.g. subcutaneous application of apomorphine), gene therapy (e.g. viral delivery of glial-derived neurotrophic factor genes), anti-inflammatory (e.g. the diabetes-approved drug Exenatide or omega-3 fatty acids), anti-oxidant (e.g. oral cannabidiol) or regenerative strategies (e.g. to promote dopaminergic neurogenesis) are being tested in clinical trials.

  1. Are there any other situations during which oxygen deficiency occurs for example chronic obstructive pulmonary disease or other examples of chronic disorders? Are there any information about risk of Parkinson’s disease onset and aforementioned chronic disorders?

Re. Thanks for another great point. Indeed, there are intriguing – but insufficiently explored - links between respiratory diseases and PD. We now mention these in an exemplary way for COPD and obstructive sleep apnea:

There are, however, indications for a significantly higher risk of people with respiratory diseases (e.g. chronic obstructive pulmonary disease COPD (Li et al., 2015) or obstructive sleep apnea (Chen et al., 2015)) to develop PD. In addition, people with PD and COPD comorbidity are at greater risk for PD-related hospitalization (Hommel et al., 2022), potentially indicating a worse disease progression in this subpopulation.

  1. In line 197-199 authors write that alpha-synuclein levels are known to change in response to hypoxia. Could you please specify the direction of these changes?

Re. We apologize for the previous omission of this crucial information: alpha-synuclein levels increase in hypoxia, which further supports the interplay of alpha-synuclein pathology and hypoxia.

Interestingly, alpha-synuclein levels are known to increase in response to hypoxia, as demonstrated in cellular models [54] and mouse brain [55].

  1. Summing up can we assume that stroke may be the cause of the Parkinson’s disease onset? Is there any relationship between Parkinson’s disease onset/pathology and vascular risk factors, diabetes or hypertension?

Re. Thank you for this thought-provoking statement. There are associations between stroke and vascular risk factors with PD, see for example:

  • Komici 2021. Diabetes Mellitus and Parkinson’s Disease: A Systematic Review and Meta-Analyses
  • Ng 2021. Case-control study of hypertension and Parkinson’s disease
  • Chen 2019. Association between Hypertension and the Risk of Parkinson’s Disease: A Meta-Analysis of Analytical Studies

However, whether stroke can indeed cause PD and how vascular risk factors relate to PD development and progression is unclear. In our opinion, based on the evidence collected in this review, stroke or other events damaging blood delivery systems with effects on brain areas that are vulnerable in PD (including the SN) probably represent triggers for PD. However, there are several points that render this a complicated problem to investigate and remains speculative at this point. First, the long sub-clinical disease progression of PD makes the establishment of causation difficult (might be interesting to study this in rodents, but these results would be tricky to generalize to humans, since rodents most likely do not naturally develop PD and PD progression in humans usually takes many years longer than the usual lifespan of a mouse or rats until occurrence of symptoms). Second, related epidemiological studies are highly susceptible to confounders, particularly such related to age (PD like most cardiovascular risk factors strongly correlate with age). Third, the location and consequences of vascular damage in the brain will likely trigger different pathologies, depending on which circuits are primarily affected; disentangling cause and consequence is not trivial.

We added in the text (chapter 3):

Similarly, stroke and PD appear to both increase the risk of the respective other, with a pooled odds ratio of PP postmortem brains exhibiting stroke pathology amounting to 1.86, as assessed by a recent metanalysis (Liu et al., 2020). Although causality remains to be established, this could indicate that diseases or events promoting hypoxia might facilitate or accelerate the development of PD.

  1. Are there any natural defense mechanism against hypoxia that could be supported to prevent neurodegenerative disorders including Parkinson’s disease?

Re. Yes, these are especially the molecular and systemic responses initiated for example in high-altitude acclimatization. From studies e.g. on mild cognitive impairment, it would seem that also controlled intermittent hypoxia protocols (and maybe certain breathing exercises) can induce natural defense mechanisms that appear to counteract cognitive decline. The mentioned ongoing clinical trial on PD might provide some specific information on the potential in PD (with the discussed caveats). We now mention these defense mechanisms in the new figure 1 and added in the text (end of chapter 7):

In summary, there is evidence that cellular and systemic natural defense mechanisms to hypoxia can be activated both by pharmacological (e.g. modulation of the HIF-pathways) or non-pharmacological ways (e.g. controlled exposure to inspirational hypoxia). These defense mechanisms (see Figure 1) importantly include antioxidant- and anti-inflammatory mechanisms and the induction of (largely HIF-mediated) protective mechanisms to maintain energy availability and cellular homeostasis, by improvements of mitochondrial resilience, increased flexibility of substrate utilization and switching to alternative metabolic pathways (primarily glycolysis), the optimization of oxygen delivery, as well as by bolstering proteolytic and autophagic capacities [21, 36], all neuroprotective mechanisms that may counteract PD neuropathology. Conversely, uncontrolled (both continuous and intermittent) hypoxia is detrimental for cells and organs, if the activation of the defense mechanisms is insufficient, either because the hypoxia is too severe, or the defense capacities too weak. Conversely, uncontrolled (both continuous and intermittent) hypoxia is detrimental for cells and organs, if the activation of the defense mechanisms is insufficient, either because the hypoxia is too severe, or the defense capacities too weak. Regarding waste disposal, the unfolded protein response (Bartoszewska and Collawn, 2020), chaperone mediated autophagy (Dohi et al., 2012; Tripathi et al., 2019) and mitophagy (Wu and Chen, 2015) are all implicated in PD pathology and are modulated by hypoxic stress.

  1. Paragraph 7: Could you precise what molecular mechanisms may underlie the beneficial hypoxia, preserving brain metabolism and neuroprotection? Could you support it with scheme presenting detrimental and beneficial roles of oxygen deficits?

Re. Thank you for this suggestion; we now provide a new figure (figure 1) outlining some of the hypothesized mechanisms for both beneficial and detrimental effects of hypoxia in PD.